

# Assessment of agro-morphological, physiological and yield traits diversity among tropical rice

Naqeebullah Kakar[1], Raju Bheemanahalli[1], Salah Jumaa[1,2],
Edilberto Redoña[3], Marilyn L. Warburton[4] and Kambham R. Reddy[1]

[1] Department of Plant and Soil Sciences, Mississippi State University, Mississippi State, MS, United States
[2] Field Crops Department, Tikrit University, Tikrit, Iraq
[3] Delta Research and Extension Center, Mississippi State University, Stoneville, MS, United States
[4] Corn Host Plant Resistance Research Unit, Crop Science Research Laboratory, USDA-ARS, Mississippi State, MS, United States

Corresponding author
Kambham R. Reddy,
krreddy@pss.msstate.edu

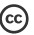

## ABSTRACT

Rice (*Oryza sativa* L.) is an essential staple food crop, but the per acre average rice yield is less than its substantial potential in many countries. Rice breeders and growers would benefit from a robust genotypes with better morpho-physiological and yield-related traits. Here, seventy-four new rice genotypes were phenotyped over two years for their gas exchange and yield potential-related traits under Mississippi rice-growing conditions. A wide range of variability was observed among genotypes for all measured traits. Detailed phenotyping of rice genotypes revealed two key relationships that function together to contribute to yield potential under the southern US climate. The first one, grain yield, grain number, and spikelet fertility, showed considerable correlation (r = 0.45 to 0.79, $p < 0.001$) to harvest index. Conversely, days to anthesis had a high and negative correlation with harvest index (r = −0.79, $p < 0.001$), which suggests that selection for short duration genotypes with efficient partitioning could improve the yields under southern US climatic conditions. Additive response index revealed a higher positive association with yield traits ($R^2 = 0.59$) than physiological ($R^2 = 0.28$) and morphological traits ($R^2 = 0.21$). Compared with the commercial genotype Rex, 21.6% and 47.3% of the rice genotypes had a higher gas exchange and yield response scores. IR08A172, IR07K142 and IR07F287 were ranked as high performers in physiological and yield response indices. Our study highlights that selection for short-duration yield-related traits with efficient sink capacity traits is desirable for future breeding programs.

## INTRODUCTION

Rice (*Oryza sativa* L.) is an important food crop that feeds more than half of the world's population. It is consumed as a staple food in the rice-producing areas of Asia, Africa, and South America. Global rice production has increased considerably from 1.86 to 4.66 t ha$^{-1}$

over the past six decades (http://www.fao.org/faostat/en/#data/QC/visualize). Currently, rice is being grown in variable climate conditions in more than 100 countries, on an estimated area of about 163 million ha with 481 million metric tons of annual production. However, the current level of rice productivity is not enough to feed the rapidly increasing global population (*Foley et al., 2011*; *Lesk, Rowhani & Ramankutty, 2016*) and need to be improved quantitatively and qualitatively to ensure global food security (https://www.yieldgap.org/web/guest/yieldgaps). On the other hand, global rice cropland areas witness yield stagnation or collapse up to 35% (*Ray et al., 2012*). With the global population multiplying (projected to reach 10.9 billion before 2100), the rice yield gains stagnation in recent years is predicted to have profound implications on the food chain (*Ray et al., 2012*). There is a need to break these slowing yield trends to minimize negative impacts on global food production.

Morphological and yield-associated traits have frequently been used as a criterion to evaluate phenotypic variability and as the basis for the enhancement of rice-yield potential. In rice, the yield is associated with several component traits (*Li et al., 2014*), including panicle number per unit area, filled grains per panicle, and 1,000-grain weight (*Yoshida, Satake & Mackill, 1981*; *Sakamoto & Matsuoka, 2008*; *Mohammadi et al., 2010*; *Xing & Zhang, 2010*; *Huang et al., 2013*; *Li et al., 2019*), and with other traits such as plant height, tillering ability, canopy architecture (*Huang et al., 2013*), total biomass production (*Long et al., 2006*) and photosynthetic efficiency (*Zhu, Long & Ort, 2008*). Genetic variation in gas exchange and pigment traits have been explored and used in different breeding programs (*Zhao et al., 2010*; *Kondamudi et al., 2016*; *Haritha et al., 2017*). Photosynthesis has been the precondition for a successful breeding program to increase photo-assimilate production in high-yielding genotypes (*Haritha et al., 2017*). For example, genotypes with slower Rubisco degradation or stay green trait resulted in a longer duration of photo-assimilation and higher yield (*Shin et al., 2020*). Among rice different subspecies, *indica* subspecies seems to have a rapid life cycle due to earlier senescence traits (*Abdelkhalik et al., 2005*). Direct selection of yield-related traits, which are easier to measure precisely than yield itself, have been used as an effective strategy for yield improvement (*Mehetre et al., 1994*; *Samonte, Wilson & McClung, 1998*; *Kumar et al., 2014*). Thus, improving yield-related physiological or correlated traits has become a primary objective for rice improvement over the years under non-stress conditions (*Dutta, Dutta & Borua, 2013*).

In the United States, rice is mainly grown in two distinct regions, i.e., the US Mid-south and the Sacramento Valley in California. Compared with global rice production, rice yield in the US increased several folds (4.5–8.53 t ha$^{-1}$ http://ricestat.irri.org:8080/wrs) over the last two decades due to the utilization of different germplasm pools and improved management practices (*McKenzie et al., 2015*). However, the increasing frequency of storms and floods, temperature extremes, drought events, and rapid urbanization coupled with increasing population are still significant constraints to ensuring food security (*Du et al., 2015*). Thus, raising yield potential under non-stress conditions has been a fundamental goal of rice breeding efforts (*Peng et al., 2008*; *Guan et al., 2010*). Rice producers in the USA prefer short- to medium-duration genotypes, which enable mechanized harvesting. Despite, the increase in demand for short-duration rice genotypes,

physiological and genetic improvement has been slow (*Won et al., 2020*). The lack of stable, high-yielding short-duration varieties would soon becoming a major limitation in achieving maximum yield potential in the USA rice production system. Because of the narrow planting window in the US, late-planted rice tends to be lower yielding as it experiences long daylength along with other in-season drought or increased disease pressure and fall frost during crop maturity. On the other hand, spring cold impacts seedling emergence if rice is planted too early. Due to variability in challenges at the start and end of the growing season, there is a need to identify suitable rice genotypes for a short growing window. Thus, developing or identifying high-yielding, short-duration rice genotypes with efficient source-sink capacity or high harvest index is critical.

Understanding the relationship between growth and developmental, physiological, and yield-related traits and other contributing characters helps maximum breeding gain from a selection. The additive vigor indices are a reliable method of assessing the growth rate and development of genotype performance under fluctuating environmental conditions (*Reddy et al., 2021*). Here, we show how the additive response index can help identify relationships between morpho-physiological and yield-related traits of rice genotypes under non-stress conditions. We phenotyped a panel of 74 tropical rice genotypes over two years for gas exchange, yield, and yield-related traits to identify short-duration high-yielding rice for the mid-southern US rice growing climate. The specific objectives of this study are to (1) evaluate variation in gas exchange and pigments traits at the reproductive stage, (2) assess the phenotypic performance of new rice genotypes for yield potential traits, (3) develop a method to determine additive index variability among rice genotypes, and (4) identify high yielding short-duration rice genotype/s with greater partitioning efficiency.

## MATERIALS & METHODS

### Germplasm, experimental setup and growth conditions

Plant material comprised of 74 rice breeding lines (mainly belonging to *indica* subspecies but including two local tropical *japonica* cultivars from the Mid-South US for comparison) obtained from the International Rice Research Institute (IRRI) in Los Banos, Philippines (Table S1). An experiment was conducted at the Environmental Plant Physiology Laboratory, Mississippi State University, Mississippi State (lat. 33° 28′ N, long. 88° 47′ W) in summer 2016 (planted on April 29) and 2017 (planted on April 24), in a randomized complete block design (RCBD) with five replications and 74 rice breeding lines. A total of 370 PVC pots (5 pots per genotype) with dimensions of 6″ diameter by 24″ high was filled with sand as a soil medium and arranged in the natural sunlight from sowing. Initially, six seeds were sown per pot for each genotype, thinned to one seedling per pot for each genotype 12 days after sowing (DAS). A drip irrigation system was installed to irrigate the experiment with freshwater until seedling emergence; once seedlings emerged, they were irrigated with Hoagland nutrition Solution three times daily (8:00 am, 12:00 pm, 5:00 pm) to avoid any confounding effect of drought stress and nutrient deficiency. Weather data for the growing seasons were obtained from the National Water and Climate Center (https://www.wcc.nrcs.usda.gov/). Average ambient minimum and maximum air temperature during the cropping window were 19.9 °C (SD ± 4.5) and 30.8 °C

(SD ± 3.9) in 2016, and 18.5 °C (SD ± 3.8) and 28.9 °C (SD ± 2.4) in 2017. The corresponding relative humidity (RH) were 85.9% (SD ± 3.4) and 88.4% (SD ± 4.6) in 2016 and 2017.

## Data collection

### Gas exchange and pigments

Physiological traits including net photosynthesis (Pn, $\mu$mol m$^{-2}$ s$^{-1}$), stomatal conductance (Cond, mol m$^{-2}$ s$^{-1}$), leaf transpiration rate (Tr, mmol H$_2$O m$^{-2}$ s$^{-1}$), water use efficiency (WUE, mmol CO$_2$ mol$^{-1}$ H$_2$O), chlorophyll fluorescence (Fv'/Fm') and electron transport rate (ETR, $\mu$mol m$^{-2}$ s$^{-1}$) were measured using LI-6400 portable photosynthesis system (LI-COR Inc., Lincoln, Nebraska, USA). A freshly expanded penultimate leaf from each plant was used for taking physiological measurements by using LI-6400. Before taking measurements, the instrument (LI-6400) was set to for photosynthetic photon flux density (PPFD) (1,500 $\mu$mol m$^{-2}$ s$^{-1}$) based on 640-02 LED light source, temperature (30 °C) and CO$_2$ concentration (400 ambient CO$_2$ level in the outdoors); whereas instantaneous water-use efficiency (iWUE) was calculated as the ratio of photosynthesis (Pn) to transpiration rate (Tr) per plant.

Chlorophyll content (Chl) and carotenoids (car) were measured for all the genotypes by taking the fresh penultimate leaves at the vegetative growth stage (50–60 DAS) of the individual plants from all five replications. Five leaf discs (2.0 cm$^2$ each) were carefully attained from the mid-blade of the fresh leaf without including mid-vein and set in a vial (five-ml) with five-ml dimethyl sulphoxide (DMSO). The vials were then incubated in a dark chamber at room temperature for 24 h to affluence the complete extraction of chlorophyll into the solution. Bio-Rad ultraviolet/VIS spectrophotometer (Bio-Rad Laboratories, Hercules, CA, USA) was used to measure the absorbance of extracts in the polypropylene microtiter plates and estimate the concentrations of carotenoid contents and total chlorophyll (chla+b) as described by *Chapple et al. (1992)*. The absorbance values at 470, 648 and 663 nm were used to calculate the concentrations of total chlorophyll contents and carotenoids (*Lichtenthaler, 1987*).

### Agro-morphological traits

Days to 50% seedling emergence (50%, E) was manually recorded for each genotype when more than three seeds emerged out of the total six seeds per pot. Agro-morphological traits such as tillers number (TN, no. plant$^{-1}$) and plant height (PH, cm plant$^{-1}$) were measured manually before flowering at the vegetative stage (50–60 days after sowing), and grain filling stage (100–110 DAS). Days to heading or anthesis (DH) refers to the number of days from sowing to the first panicle emergence in each plant. For dry shoot weight (SHW, g plant$^{-1}$), plants were harvested and stored in a dryer at 70 °C for at least 72 h or until thoroughly dried, and dry weights were recorded for each genotype using a bench weighing balance (Adventure$^{TM}$, OHAUS Corporation, NJ, USA).

### Yield and yield-related parameters

Several yield-related traits were measured, including panicle length plant$^{-1}$ (PL), spikelet numbers panicle$^{-1}$ (SPN), number of grains panicle$^{-1}$ (GN), grain weight panicle$^{-1}$ (GW), filled and unfilled grain percentage (GF), and total grain yield plant$^{-1}$ (GY). Panicle

length (PL, cm) was manually measured using a ruler (ACE Hardware, Metal ruler) as the distance (cm) from the panicle neck node base to the last spikelet's tip of the same panicle. Five panicles were collected from each rice plant as representative samples, labeled, and air-dried in the laboratory for approximately two weeks to measure SPN, GN, and GW. For shoot dry weight (SHW, g plant$^{-1}$), leaves and stems of individual plants were harvested, dried, and weighted. Total grain yield (GY, g plant$^{-1}$) was measured as the sum of the entire grains from all the individual plants of the same genotype in five replications. The spikelet fertility in the percentage (SF, %) was measured after separating the filled grains from the unfilled grain in all genotypes. Harvest index (HI) was calculated as a ratio of grain yield and biological yield (*Amanullah & Inamullah, 2016*).

## Statistical analysis

Statistical analyses were carried out using RStudio 3.6.1 (https://www.R-project.org/; *R Core Team, 2020*). ANOVA was performed to estimate the significance of genetic variability among the 74 rice genotypes for all measured traits concerning year, cultivars, and year x cultivars interaction (Table 1). In this study, ANOVA revealed no influence of years on most of the traits measured except for plant height, tiller number, day to heading, and panicle length. Therefore, data generated from two years were combined to explore genetic variation in gas exchange, growth, and development associated with yield potential. Principal component analysis, a multidimensional preference analysis technique that identifies parameters that best describe the variations among the genotypes was used to separate rice genotypes into four groups using XLSTAT. A correlation matrix was developed using the library ("PerformanceAnalytics"). The jitter plots for variables were created using the library ("ggpubr").The individual response index (IRI) was determined as the ratio between the value of each rice genotype ($V_x$) and the modern high-yielding rice genotype (Rex) value ($V_m$), as shown in Eq. (1).

$$IRI = V_x/V_m \tag{1}$$

For the additive response index (ARI), all the individual indices of response traits for each rice genotype at vegetative, grain filling stages, and final harvest were summed (Eq. (2)).

$$
\begin{aligned}
ARI = {} & \left(\frac{Pnx}{Pnm}\right) + \left(\frac{Condx}{Condm}\right) + \left(\frac{Trx}{Trm}\right) + \left(\frac{WUEx}{WUEm}\right) + \left(\frac{Fv'/Fm'x}{Fv'/Fm'm}\right) \\
& + \left(\frac{ETRx}{ETRm}\right) + \left(\frac{Chlx}{Chlm}\right) + \left(\frac{Carox}{Carom}\right) + \left(\frac{DEx}{DEm}\right) + \left(\frac{DHx}{DHm}\right) + \left(\frac{PHx}{PHm}\right) \\
& + \left(\frac{TNx}{TNm}\right) + \left(\frac{SHWx}{SHWm}\right) + \left(\frac{PLx}{PLm}\right) + \left(\frac{SPNx}{SPNm}\right) + \left(\frac{GWx}{GWm}\right) + \left(\frac{GNx}{GNm}\right) \\
& + \left(\frac{SFx}{SFm}\right) + \left(\frac{GYx}{GYm}\right) + \left(\frac{HIx}{HIm}\right)
\end{aligned}
\tag{2}
$$

The ARI (on average ARI of 74 genotypes = 21.5) and standard deviation (SD = 2) were then used to classify the 74 rice genotypes into potential highly desirable ones (high), and

**Table 1  Analysis of variance (ANOVA) across 74 rice genotypes for morpho-physiological and yield-related traits measured during vegetative and grain filling stages for two years.**

| Trait | Trait acronym/unit | Two-way ANOVA | | | Minimum | Maximum | Mean | SD |
|---|---|---|---|---|---|---|---|---|
| | | Cultivar (C) | Year (Y) | C × Y | | | | |
| **Physiological traits** | | | | | | | | |
| Net photosynthesis | Pn, $\mu mol\ m^{-2}\ s^{-1}$ | *** | NS | NS | 20.80 | 36.76 | 29.89 | 3.47 |
| Stomata conductance | Cond, $mol\ m^{-2}\ s^{-1}$ | *** | NS | NS | 0.59 | 2.01 | 1.20 | 0.28 |
| Transpiration rate | Tr, $mmol\ H_2O\ m^{-2}\ s^{-1}$ | *** | NS | NS | 9.19 | 13.76 | 11.34 | 1.03 |
| Instantaneous Water-Use Efficiency | iWUE, $mmol\ CO_2\ mol^{-1}\ H_2O$ | *** | NS | NS | 2.19 | 3.37 | 2.68 | 0.20 |
| Chlorophyll fluorescence | Fv'/Fm' | *** | NS | NS | 0.42 | 0.86 | 0.52 | 0.05 |
| Electron transport rate | ETR, $\mu mol\ m^{-2}\ s^{-1}$ | *** | NS | NS | 126.75 | 226.18 | 157.45 | 15.86 |
| Total chlorophyll | CHL, $\mu g\ cm^{-2}$ | *** | NS | NS | 28.35 | 49.19 | 37.84 | 5.22 |
| Carotenes | Caro, $\mu g\ cm^{-2}$ | *** | NS | NS | 4.23 | 8.53 | 5.89 | 0.87 |
| **Morphological traits** | | | | | | | | |
| Days to 50% emergence | DE, days | *** | NS | NS | 8.0 | 12.0 | 10.16 | 0.92 |
| Day to heading | DH, days | *** | *** | ** | 85.6 | 160.7 | 121.71 | 19.15 |
| Plant height | PH, $cm\ plant^{-1}$ | *** | *** | *** | 13.5 | 100.5 | 54.63 | 18.03 |
| Tiller number | TN, $no.\ plant^{-1}$ | *** | *** | *** | 49.1 | 114.8 | 79.72 | 10.63 |
| Shoot dry weight | SHW, $g\ plant^{-1}$ | *** | NS | NS | 65.5 | 377.9 | 191.07 | 67.61 |
| **Yield-related traits** | | | | | | | | |
| Panicle length | PL, cm | *** | NS | *** | 16.75 | 30.13 | 23.60 | 2.74 |
| Spikelet number | SPN, $no.\ panicle^{-1}$ | *** | NS | NS | 8.09 | 21.80 | 12.03 | 1.77 |
| Grain weight | GW, $g\ panicle^{-1}$ | *** | NS | NS | 0.67 | 3.74 | 2.12 | 0.69 |
| Spikelet fertility | SF, % | *** | * | NS | 18.38 | 95.55 | 73.23 | 16.07 |
| Grain number | GN, $no.\ plant^{-1}$ | *** | NS | NS | 60.12 | 199.42 | 126.32 | 28.83 |
| Grain yield | GY, $g\ plant^{-1}$ | ** | NS | NS | 18.80 | 127.45 | 67.64 | 27.56 |
| Harvest index | HI, plant basis | *** | NS | NS | 0.07 | 0.48 | 0.27 | 0.10 |

Note:
Significance level: $^*p < 0.05$, $^{**}p < 0.01$, $^{***}p < 0.001$; NS, non-significant. SD – standard deviation.

moderately desirable (Moderately High), undesirable (Moderately Low), and highly undesirable ones (Low) using the following equations.

$$High = genotypes\ with\ ARI\ values > 23.5 \tag{3}$$

$$Moderately\ High = genotypes\ with\ ARI\ values\ between < 23.5\ and > 21.5 \tag{4}$$

$$Moderately\ Low = genotypes\ with\ ARI\ values\ between < 21.5\ and > 19.5 \tag{5}$$

$$Low = genotypes\ with\ ARI\ values < 19.5 \tag{6}$$

The top-ranking genotypes for physiological and yield traits under well-watered conditions were identified. Graphs for the relationship between morpho-physiological and yield-related traits were plotted using Sigma Plot 13 (Systat Software Inc., San Jose, CA, USA).

## RESULTS AND DISCUSSION

### Phenotypic variation

Complex physiological and biochemical processes control plant growth and development. Each developmental phase involves various metabolic changes and is affected by diurnal factors such as light, temperature, and day length. Rice growing seasonal variation in the US is different from other more tropical regions such as Africa, India, and the Philippines due to the onset and offset cold-weather windows. This study characterized variation in 74 genotypes of *O. sativa* obtained from IRRI for morpho-physiological and yield-related traits over two years under non-stress conditions. A list of traits measured and their ranges are provided in Table 1. The variation in physiological, morphological, and yield-related traits differed significantly among genotypes ($p < 0.001$). Most of the traits were not affected ($p > 0.05$) by Year (2016 and 2017) and Genotype by Year interactions except for plant height, days to heading, and tiller number (Table 1). Due to the non-significant difference in mean average temperatures (24.4 ± 4.0 °C in 2016 and 23.4 ± 2.9 °C) and relative humidity (85.6% in 2016 and 88.4% in 2017) between years, the two years data points were combined for further analysis.

Under non-stress conditions, the average number of days to 50% emergence (when three seeds emerged out of six) varied from 8 to 12 days after planting. Emergence could be a useful trait in selecting vigorous genotypes for breeding early seedling stage heat and drought tolerance (*Reddy et al., 2021*). Most of the rice genotypes usually germinate within a week. However, different germination rates could be affected by additional factors, including soil medium and weather conditions. The growth and developmental traits, including TN, PH, and SHW, displayed significant variability. Genotypes IR49830 and IR86635 were the earliest and latest flowering, with 86 and 161 days to flowering, respectively. In the present study, the variance of gas exchange parameters, grain yield and its components in rice showed significant differences among genotypes (Table 1). A wide phenotypic variability among genotypes indicates the potential to identify donors for trait manipulation through breeding.

### Principal component and correlation analysis

Principal component analysis (PCA) was performed to explore the relationship between traits and rice genotypes under non-stress conditions (Figs. 1A–1C). The first two principal components (PC) cumulatively explained 41.7% (PC1: 25.6% and PC2: 16.2%) of the total phenotypic variation (Fig. 1A). PC1 contributed more towards yield and related traits to the separation of rice genotypes and captured the variability for harvest index (16.4%), grain weight (15.4%), grain number (12.3%), days to heading (10.1%) and grain yield (10%). PC2 explained 16.2% of the variation between genotypes, mostly contributed by physiological traits such as Pn (22.7%), Chl (17.1%), Caro (12.6%), and Tr (11.3%). PC1 showed a positive association with GY (0.7), GW (0.87), GN (0.77), HI (0.89) and SF (0.47). In contrast, the number of days to heading (−0.70) and SHW at maturity (−0.52) were negatively correlated with PC1. Gas exchange parameters such as Pn (0.83), Chl (0.82), Caro (0.62) and Tr (0.59) were positively associated with PC2 (Fig. 1B). A 2D

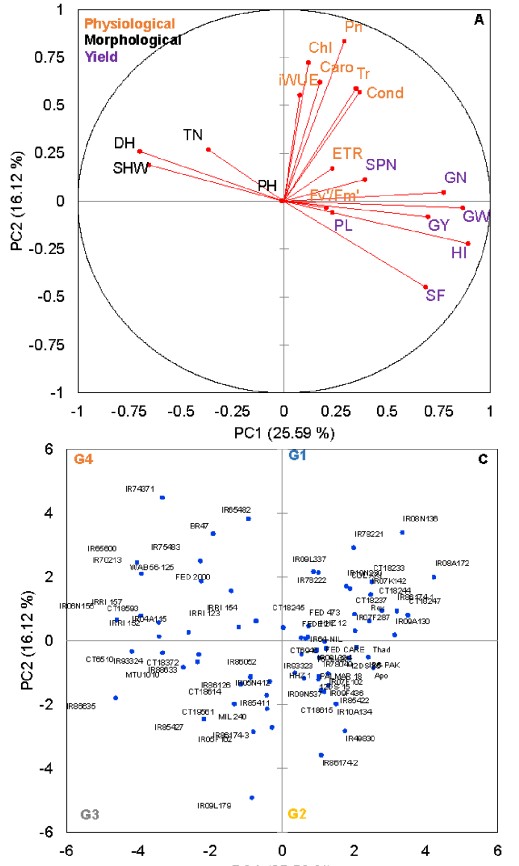

| B | PC1 | | PC2 | | PC3 | |
|---|---|---|---|---|---|---|
| Eigenvalue | 4.9 | | 3.1 | | 2.6 | |
| Variability (%) | 25.6 | | 16.1 | | 13.5 | |
| Cumulative % | 25.6 | | 41.7 | | 55.2 | |
| | Cont | Cor | Cont | Cor | Cont | Cor |
| Pn | 1.76 | 0.29 | 22.73 | 0.83 | 2.08 | −0.23 |
| Cond | 2.78 | 0.37 | 10.54 | 0.57 | 8.95 | −0.48 |
| Tr | 2.52 | 0.35 | 11.28 | 0.59 | 10.01 | −0.51 |
| WUE | 0.12 | 0.08 | 10.01 | 0.55 | 2.98 | 0.28 |
| Fv'/Fm' | 0.86 | 0.20 | 0.04 | −0.03 | 0.39 | 0.10 |
| ETR | 1.13 | 0.23 | 0.96 | 0.17 | 0.09 | 0.05 |
| Chl | 0.29 | 0.12 | 17.12 | 0.72 | 3.44 | 0.30 |
| Caro | 0.63 | 0.17 | 12.59 | 0.62 | 2.70 | 0.26 |
| DH | 10.07 | −0.70 | 2.21 | 0.26 | 0.07 | 0.04 |
| TN | 2.78 | −0.37 | 2.38 | 0.27 | 14.39 | 0.61 |
| PH | 0.00 | −0.01 | 0.00 | 0.00 | 14.47 | 0.61 |
| SHW | 8.80 | −0.65 | 1.17 | 0.19 | 15.32 | 0.63 |
| PL | 1.12 | 0.23 | 0.11 | −0.06 | 8.65 | 0.47 |
| SPN | 3.16 | 0.39 | 0.42 | 0.11 | 3.44 | 0.30 |
| GW | 15.45 | 0.87 | 0.04 | −0.03 | 1.02 | 0.16 |
| SF | 9.71 | 0.69 | 6.54 | −0.45 | 0.61 | 0.13 |
| GN | 12.34 | 0.77 | 0.07 | 0.05 | 3.36 | 0.29 |
| GY | 10.04 | 0.70 | 0.21 | −0.08 | 7.94 | 0.45 |
| HI | 16.43 | 0.89 | 1.59 | −0.22 | 0.07 | −0.04 |

**Figure 1 The principal component analysis (PCA) of the morpho-physiological and yield-related traits with the first two principal components (PC1 and PC2) in non-stress (A).** Trait labels are colored differently according to trait category (physiological traits in orange, morphological traits in black, and yield-related traits in purple) in Table 1; acronyms are given in Table 1. Contribution (Cont) of the variables (%) and correlations (Cor) between variables and factors to the principal components (B). Classification of 74 rice genotypes based on the factor scores of first (PC1) and second (PC2) principal components. G1 to G4 represents rice genotypes grouping (C).

scatter plot of factor loading values separated 74 genotypes into four different groups (Fig. 1C) A positive relationship was found between DH and SHW ($r = 0.58$, p < 0.001). At the same time, DH showed strong and negative correlations ($r = -0.44$ to $-0.79$, $p < 0.001$) with most of the yield-related variables (Fig. 2). Similarly, SHW was negatively correlated with GW ($r = -0.44$, $p < 0.001$), SF ($r = -0.39$, $p < 0.001$), and HI ($r = -0.70$, $p < 0.001$), which indicates that genotypes with a longer life span are associated with lower yield potential (Fig. 2) under the mid-south US climatic conditions. Although early flowering genotypes have a short period of vegetative growth, the reproductive and grain filling stages have been reported to be similar for rice (*Kropff, Van Laar & Matthews, 1994*). These observations indicate that selection of short-duration rice genotypes with no plateauing in yield would offer multiple advantages compared with long-duration rice genotypes in the southern US growing season.

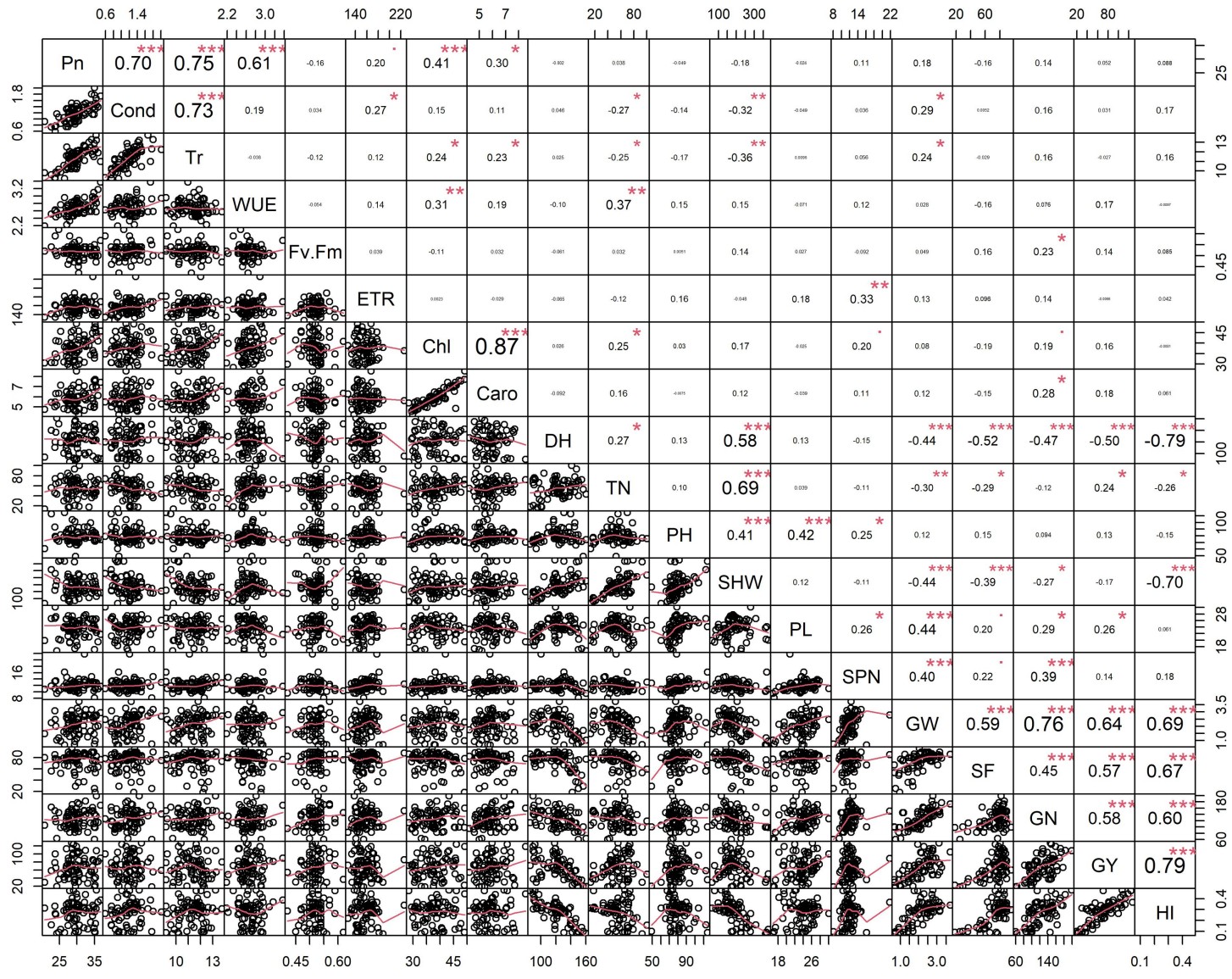

**Figure 2 Pearson's correlation coefficients among 74 rice genotypes in non-stress conditions.** Correlations values with ± indicate a strong between two traits. Asterisks (\*, \*\*, \*\*\*) indicate significance at <0.05; <0.01; <0.001, respectively. Physiological traits: Net photosynthesis (Pn), Stomata conductance (Cond), Transpiration rate (Tr), Water use efficiency (iWUE), chlorophyll fluorescence (Fv′/Fm′), Electron transport rate (ETR), Total chlorophyll (Chl) and Carotenes (Caro); Morphological traits: days to heading (DH, d), tiller number (TN, per plant), plant height (PHT, cm) and shoot dry weight (SHW, g per plant); yield-related traits: panicle length (PL, cm), spikelet number per panicle (SPN, no per panicle), grain weight per panicle (GW, g per panicle), spikelet fertility (SF, %), grain number per plant (GN, per plant), grain yield (GY, g per plant) and harvest index (HI, plant basis) traits values among 74 rice genotypes in non-stress conditions.

Rice genotypes were differentiated into four groups, separating high and low performers and other combinations based on the factor scores of PC1 and PC2. As shown in the biplot (Fig. 1C), the vector's direction and angle represent the relationships among traits. The dispersion of genotypes in the same direction as the vectors helped group them with similar physiology and yield traits. Rice genotypes with better physiology and higher-yielding traits might become an essential resource for rice breeders. As suggested by

the PCA, 74 rice genotypes were classified into four groups. Group 1 and 2 included 24 rice genotypes, and group 3 included 17, and group 4 had 15 genotypes (Fig. 1C).

## Leaf gas exchange and physiological traits

There were significant differences between PC-derived groups for gas exchange traits (Figs. 3A–3H). The mean of Pn in group 1 (IR08N136) and group 4 (IR65482) were significantly ($P < 0.05$) higher than that of group 2 (IR93323) and group 3 (CT19561). On average, genotypes from group 1 had significantly greater stomatal conductance (IR10N230, Fig. 3B), Tr (IR78221, Fig. 3C), WUE (IR08N136, Fig. 3D), Chl (IR07K142, Fig. 3G), and Caro (IR07K142, Fig. 3H) than genotypes from group 3. Further, iWUE varied significantly ($p < 0.05$) among the four groups. On average, group 1 (IR08N136 and IR09L337) and group 4 (IRRI152 and IR65482) genotypes had significantly greater iWUE compared to group 2 (IR09L324) and group 3 (CT18372). However, Fv'/Fm' and ETR were not different among groups except for ETR in group 3 (Figs. 3D–3F). Like Pn, Chl and Caro were also significantly higher in group 1 and group 4 than the other two groups (Figs. 3G, 3H). The physiological characterization of plants helps assessment of the genetic diversity within the germplasm pool. Variability in leaf chlorophyll content (*Prasad & Djanaguiraman, 2011*; *Reddy et al., 2021*), carotenoids, and relative injury (*Zafar et al., 2018*) have been previously used as screening tools to identify vigorous germplasm (*Sheshshayee et al., 2006*; *Syed, Rasmus & Matthew, 2017*; *Singh et al., 2018*). Similarly, chlorophyll and leaf area are crucial in determining the yield (*Lu et al., 1994*; *Anyia, 2004*; *Kikuchi et al., 2017*; *Won et al., 2020*), as higher densities of chlorophyll per unit leaf area have been observed in thicker leaves, enhancing the photosynthetic capabilities (*Craufurd et al., 1999*) of plants resulting in the vigorous and productive crop. Overall, our results suggested that the group 1 and 4 rice genotypes are associated with better photosynthetic efficiency and high iWUE, which can be selected for biomass improvement (*Placido et al., 2013*; *Vikram et al., 2016*).

## Agro-morphological phenotype

The 74 genotypes showed significant variability in days to heading (DH), which is a useful trait because early heading lines mature earlier (*Won et al., 2020*). Sixteen of the 74 genotypes flowered earlier than control entry Rex. On average, DH was the longest in group 4 (142 d) and group 3 (125 d) genotypes, which may not be suitable for a short growing season. Days to heading was shortest in group 2 (IR49830, 85.3 d) and group 1 (IR08A172, 89. 6 d), which were 12 and 54.4 d earlier than Rex and the average of group 4 genotypes, indicating that these genotypes may be photoperiod insensitive (Fig. 4A). Greater biomass at flowering coupled with lack of photoperiod sensitivity could boost yield in US mid-south environments, and the negative correlation between DH and yield suggests opportunities to select high-yielding genotypes from short-duration genotypes. Similarly, the mean values of total tiller number (TN) in group 4 were significantly greater than in group 2 and 1 by 34.5 and 18.5%, respectively (Fig. 4B). While there was significant genetic variability in PH, it did not differ between PCA-derived groups (Fig. 4C), indicating that the DH and TN traits interacted less with PH in this population.

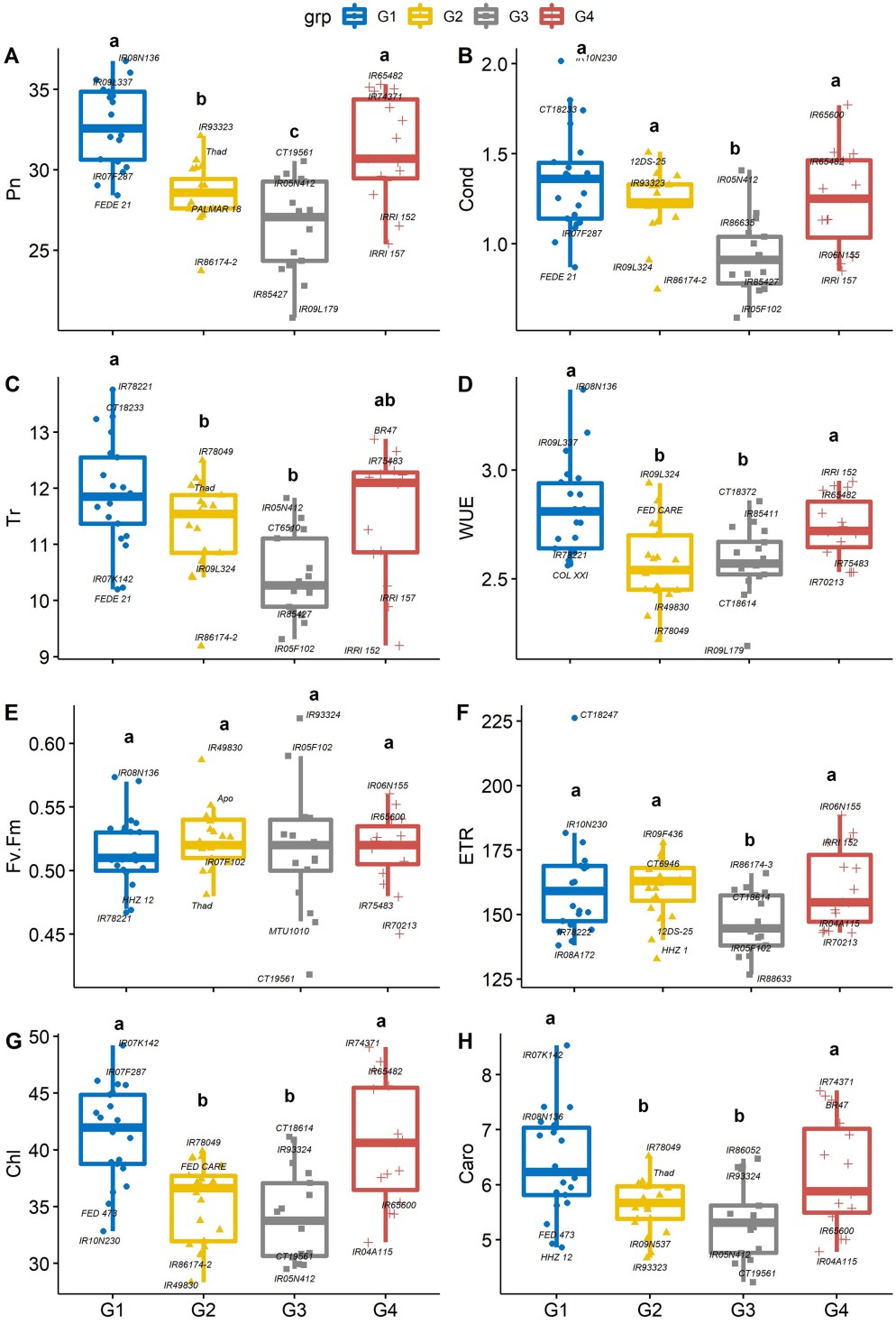

**Figure 3 Boxplots showing the differences in physiological traits.** Boxplot showing the differences in physiological traits such as (A) net photosynthesis (Pn, µmol m$^{-2}$ s$^{-1}$), (B) stomatal conductance (Cond, mol m$^{-2}$ s$^{-1}$), (C) leaf transpiration rate (Tr, mmol H$_2$O m$^{-2}$ s$^{-1}$), (D) water use efficiency (WUE, mmol CO$_2$ mol$^{-1}$ H$_2$O), (E) chlorophyll fluorescence (Fv'/Fm'), (F) electron transport rate (ETR, µmol m$^{-2}$ s$^{-1}$), (G) Chlorophyll content (Chl) and (H) carotenoids (Caro) values among four groups of rice under non-stress conditions. The four groups (G1–G4) represent rice genotype grouping based on the first and second principal components (see Fig. 1B). Different letters on the boxplot indicate a significant difference among the groups ($p < 0.05$).               

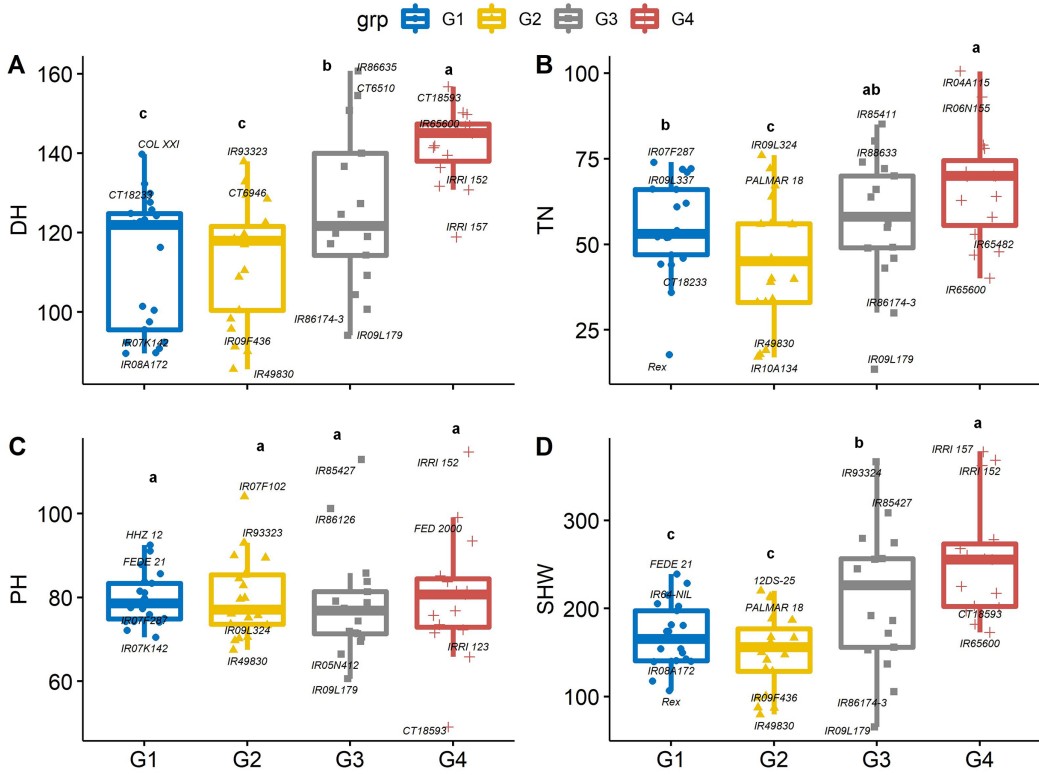

**Figure 4 Boxplots showing the differences in (A) days to heading (DH, d), (B) tiller number (TN, per plant), (C) plant height (PHT, cm) and (D) shoot dry weight (SHW, g per plant) traits values among four groups of rice genotypes under non-stress conditions.** The four groups (G1–G4) represent rice genotype grouping based on the first and second principal components (see Fig. 1B). Different letters on the boxplot indicate a significant difference among the groups ($p < 0.05$).

Genotypes with earlier flowering (i.e., group 1 and 2) also had 37.4% lower mean dry shoot weight (SHW) as compared with group 4 (Fig. 4D). Differences in shoot dry weights are related to other growth and developmental traits, including PH and TN. Genotype IRRI152 had both the highest SHW and greatest plant height, and genotype IR09L179 had the lowest SHW and least number of tillers among all the genotypes. Since similar environmental conditions were observed in both years, the differences in growth and developmental traits, particularly PH and TN, maybe due to the genetic variability among the genotypes. This can be exploited in breeding programs to screen for high-yielding cultivars. At a given growth stage, these parameters are shown to be heritable and stable for evaluating rice genotypes (*Peng et al., 2008*). Taller rice plants compete with weeds better than shorter plants, and grain yield increases quadratically with increasing plant height (*Fageria, Castro & Baligar, 2004*) up to a point. Taller plants run a risk of lodging (*Kato et al., 2019*), and a small increase in grain yield has been related to reduced plant height (*Evans, Visperas & Vergara, 1984*). Tiller number is crucial under biotic and abiotic stresses due to compensation processes. High tillering capacity is associated with the maximum use of space and resources; thus, this trait significantly affects total grain yield.

Mainly, genotypes with a higher number of effective tillers per plant produce higher rice grain yields (*Dutta, Dutta & Borua, 2013*).

## Yield potential of long duration rice genotypes under non-stress conditions

Yield and yield-related traits are a result of interactive physiological and biochemical processes during the crop cycle. Grain yield is determined by panicle number, spikelets per panicle, spikelet fertility, and individual grain weight (Fig. 5). Genotype IR65600 produced the fewest panicles and spikelets panicle$^{-1}$ and had the lowest grain yield among all rice genotypes (Table S1). Overall genotypes, 73% of the panicles had filled grains, with genotype HHZ1 containing as high as 96% and genotype IR74371 as low as 18% filled grains. Spikelet fertility was significantly affected by the year; more filled grains were observed in the first year with an average of 75% of the grains filled as compared to the second year with an average of 72% filled grains. Average panicle length, number of spikelet panicle$^{-1}$, number of grain plant$^{-1}$, and grain weight panicle$^{-1}$ were 23.6 cm, 12.0 no/panicle, 126.3 no/plant and 2.12 g, respectively (Table 1).

Panicles per plant and the number of spikelet numbers on each panicle play an essential role in increasing or decreasing the total grain yield. Genotype IR65600 produced the fewest spikelets panicle$^{-1}$ and had the lowest grain yield among all the rice genotypes. On average, groups 1 and 2 had significantly greater grain weight, spikelet fertility, grain number, grain yield, and harvest index than group 4 (Figs. 5C–5G). Significant differences were recorded for average grain yield across groups ($p < 0.05$), and group 1 and 2 genotypes were the highest yieldings (86.4 and 78.4 g plant$^{-1}$, respectively). In contrast, group 4 recorded the lowest yields (39.6 g plant$^{-1}$). The mean of group 1 and group 2 genotypes recorded the highest harvest index (0.34), and the group 4 genotypes recorded the lowest HI (0.14). The bottom 10% of the HI genotypes were mostly long-duration genotypes which produced an average of 164.5 spikelets per panicle, which is 57.4% greater than the top 10% of high HI genotypes. Although spikelet number per panicle is correlated with yield, numerous spikelets per panicle might have to fight for nonstructural carbohydrates or concurrent photo assimilation during grain filling (*Peng et al., 2008*; *Won et al., 2020*). This becomes a bottleneck in high-yielding genotypes (*Kikuchi et al., 2017*; *Fabre et al., 2020*). An alternative route to overcome this shortfall would be short-duration plants with a better sink capacity (*White et al., 2016*).

The degree of correlation among the traits is crucial, especially for a complex trait such as yield (*Guan et al., 2010*; *Akinwale et al., 2011*). Under non-stress conditions, yield component traits (GW, SF, GN and GY) are correlated (r = 0.45–0.79, $p < 0.001$) to harvest index, implying that selection for any of these could significantly improve yield. However, days to heading (anthesis) was negative correlated with GW (r = −0.44, $p < 0.001$), SF (r = −0.52, $p < 0.001$), GN (r = −0.47, $p < 0.001$), GY (r = −0.50, $p < 0.001$) and HI (0 = −0.79, $p < 0.001$). While late-flowering or maturing genotypes could have a large grain filling window, resulting in higher grain yields by pushing extra photo-assimilates to the sink (grain) under optimum conditions. Mississippi growing conditions do not appear to be optimum due to large variation in day-length compared to

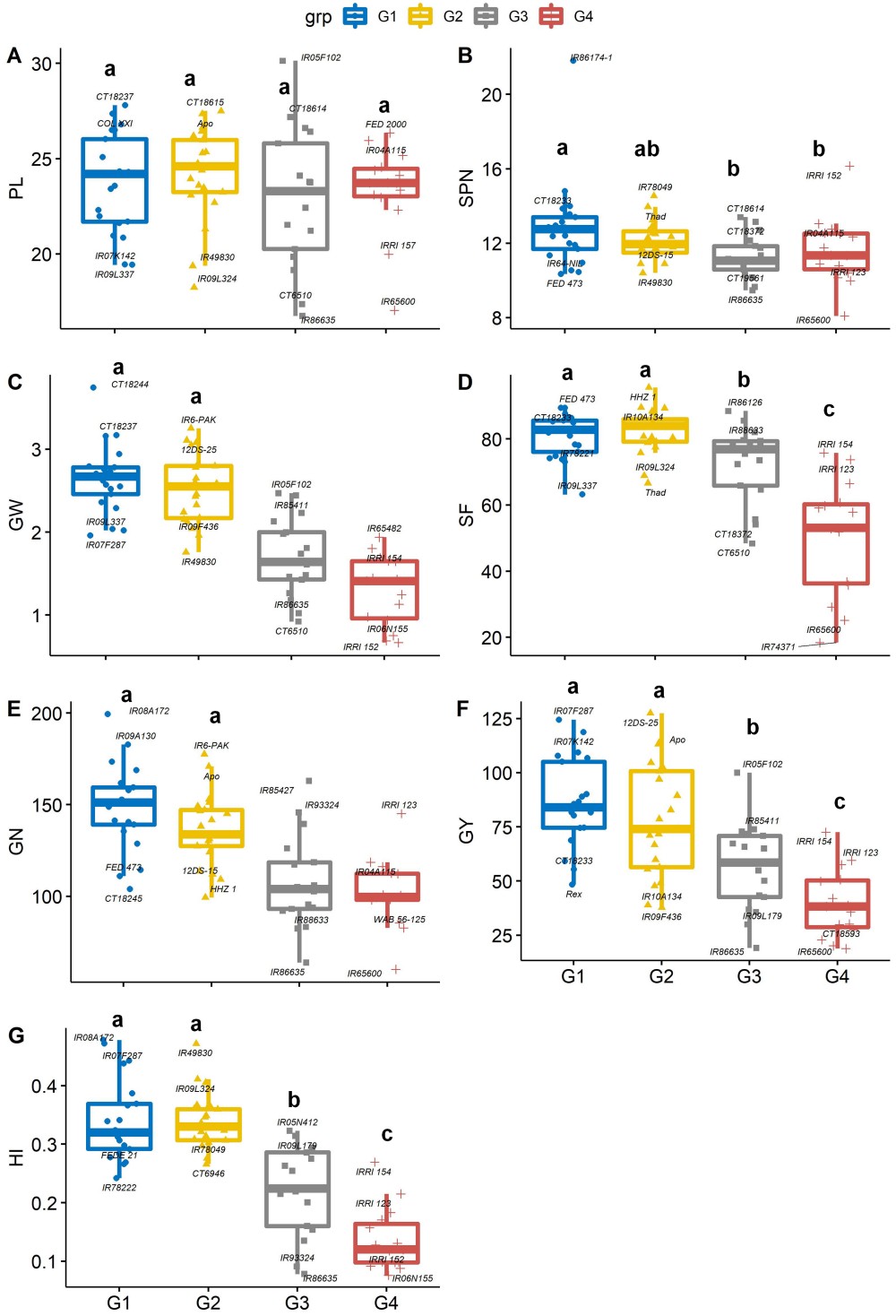

**Figure 5 Boxplots showing the differences in yield and yield components traits.** Phenotypic distribution of (A) panicle length (cm), (B) Spikelet number (SPN, no. panicle$^{-1}$), (C) grain weight (GW, g panicle$^{-1}$), (D) spikelet fertility (SF, %), (E) grain number (GN, no. plant$^{-1}$), (F) grain yield (GY, g plant$^{-1}$) and (G) Harvest index (HI) among four groups of rice genotypes under non-stress conditions. The four groups (G1–G4) represent rice genotype grouping based on the first and second principal components (see Fig. 1B). Different letters on the boxplot indicate a significant difference among the groups ($p < 0.05$).

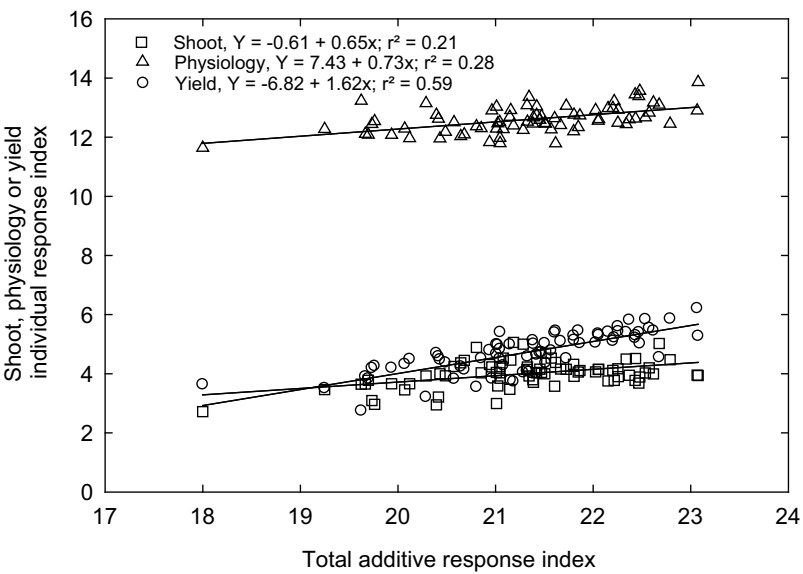

**Figure 6 The relationship between total additive response index and shoot (*square*), physiological (*triangle*) and yield (*open circle*) individual response index of 74 rice genotypes.**

tropical regions (Philippines, India, Africa, and other tropical and subtropical regions), and late flowering/maturing genotypes are exposed to late-season cold weather during grain filling, which also induces yield losses.

## Variability assessment based on physiological, growth, and yield responses index

Rex is a genotype (variety) popular in Mississippi because of adaptability and grain traits (https://www.mafes.msstate.edu/publications/information-bulletins/ib0548.pdf). Thus, the individual response index was calculated for all measured traits compared to the control genotype "Rex" (see Materials and methods). Correlations between agro-morphological, physiological, and yield-related traits with additive response indices are shown in Fig. 6, with the coefficient of determination ($R^2$) values, which give the percentage of total differences or additive response index described by each independent variable. An overall high linear positive correlation was observed between the yield-related traits ($R^2 = 0.59$) and the additive response index (ARI). In contrast, weak correlations were observed between ARI and growth and developmental traits ($R^2 = 0.21$) and physiological traits ($R^2 = 0.28$) for the selected rice genotypes (Fig. 6). A similar response index has been used as selection criteria for phenotyping cereals, legumes, and horticulture crops for abiotic stress tolerance, such as cold tolerance (*Wijewardana et al., 2015*), low or high-temperature tolerance (*Jumaa et al., 2020*; *Reddy et al., 2021*), drought tolerance (*Lone et al., 2019*; *Singh et al., 2018*), and salt tolerance (*Kakar et al., 2019*). The ARI values of all the genotypes and their standard deviations were further used to classify rice genotypes into four response groups, including low (and undesirable; 16.2% of the

**Table 2 Classification of 74 tropical rice genotypes based on the additive response index of morpho-physiological and yield-related traits measured under non-stress conditions.**

| High (n = 15) | | Moderately high (n = 28) | | Moderately low (n = 19) | | Low (n = 12) | |
|---|---|---|---|---|---|---|---|
| (23.45–24.16) | | (21.62–23.24) | | (19.71–21.28) | | (14.92–19.31) | |
| 12DS-25 | 24.16 | COL XXI | 23.24 | IR75483 | 21.28 | CT19561 | 19.31 |
| IR08A172 | 24.12 | IR85411 | 23.24 | IR93323 | 21.25 | Rex | 19.00 |
| CT18244 | 24.10 | IR86174-1 | 23.12 | MTU1010 | 21.12 | IR85422 | 18.91 |
| IR07K142 | 24.10 | IR06N155 | 22.95 | IR70213 | 21.09 | Thad | 18.66 |
| PALMAR 18 | 24.08 | CT18247 | 22.95 | IR65482 | 21.09 | CT18593 | 18.61 |
| IR64-NIL | 23.96 | IR09N537 | 22.87 | IR86174-2 | 21.08 | IR86635 | 17.82 |
| IR08N136 | 23.85 | IR78221 | 22.79 | 75-1-127 | 21.05 | IR49830 | 17.81 |
| IR07F287 | 23.67 | IR78222 | 22.75 | 12DS-15 | 21.03 | IR86174-3 | 17.66 |
| IR09A130 | 23.60 | IRRI 152 | 22.73 | IR6-PAK | 20.72 | IR10A134 | 17.64 |
| IR04A115 | 23.56 | IR09L324 | 22.68 | CT6946 | 20.56 | IR09F436 | 17.56 |
| FEDE 21 | 23.54 | CT18615 | 22.62 | IR86052 | 20.49 | IR65600 | 17.53 |
| Apo | 23.51 | IRRI 157 | 22.58 | IR74371 | 20.44 | IR09L179 | 14.92 |
| FED 2000 | 23.50 | IRRI 154 | 22.57 | HHZ 1 | 20.33 | | |
| IR05F102 | 23.50 | IR07F102 | 22.55 | IR05N412 | 20.14 | | |
| FED CARE | 23.45 | HHZ 12 | 22.51 | IR78049 | 20.12 | | |
| | | CT18237 | 22.50 | WAB 56-125 | 20.02 | | |
| | | BR47 | 22.36 | CT6510 | 19.99 | | |
| | | IR09L337 | 22.29 | MIL 240 | 19.85 | | |
| | | IR93324 | 22.11 | CT18372 | 19.71 | | |
| | | IR88633 | 22.10 | | | | |
| | | CT18245 | 21.88 | | | | |
| | | FED 473 | 21.88 | | | | |
| | | IR85427 | 21.88 | | | | |
| | | IR86126 | 21.82 | | | | |
| | | IRRI 123 | 21.75 | | | | |
| | | CT18233 | 21.63 | | | | |
| | | CT18614 | 21.63 | | | | |
| | | IR10N230 | 21.62 | | | | |

genotypes), moderately low (25.7%), moderately high (32.4%), and high (highly desirable; 25.7%) total response indices (Table 2).

## Promising high yielding rice genotypes

Rice genotypes with a combination of smaller plant size and shorter crop duration compromise harvest index under optimum conditions (*Butler et al., 2005*). However, combining the same traits might positively impact stress conditions (*Blum, 2011*). Therefore, the proposed genotypes (Table 3) with early flowering and better physiology (high Pn, Gs, and iWUE) have the advantage of overcoming adverse climatic variabilities by enabling the genotype to utilize resources more efficiently during critical growth stages.

**Table 3 Recommended rice genotypes for use in high yielding environments.**

| Genotype | Pn | Cond | iWUE | Chl | DH | PH | SHW | PL | GY | HI | CRI |
|---|---|---|---|---|---|---|---|---|---|---|---|
| 12DS-25 | 28.52 | 1.51 | 2.61 | 39.26 | 118.4 | 90.1 | 220.01 | 26.2 | 127.45 | 0.36 | 24.16 |
| IR08A172 | 35.6 | 1.45 | 2.81 | 43.26 | 89.6 | 74.2 | 117.55 | 22.3 | 107.83 | 0.47 | 24.12 |
| IR07K142 | 29.86 | 1.12 | 2.89 | 49.19 | 89.7 | 70.5 | 149.45 | 19.45 | 118.72 | 0.44 | 24.10 |
| IR08N136 | 36.76 | 1.74 | 3.37 | 45.71 | 90.8 | 74.1 | 142.48 | 21.7 | 90.12 | 0.38 | 23.85 |
| IR07F287 | 29.04 | 1.01 | 2.64 | 46.08 | 92.3 | 72.1 | 139.37 | 21.98 | 124.41 | 0.47 | 23.67 |
| IR09A130 | 32.48 | 1.16 | 2.62 | 38.77 | 97.5 | 75.9 | 140.49 | 21.63 | 109.39 | 0.43 | 23.60 |
| Apo | 28.68 | 1.28 | 2.46 | 36.86 | 119.6 | 76.7 | 167.46 | 27.35 | 113.81 | 0.40 | 23.51 |
| IR86174-1 | 32.14 | 1.21 | 2.96 | 41.95 | 100.4 | 78.6 | 153.87 | 20.85 | 88.86 | 0.36 | 23.12 |
| CT18247 | 34.60 | 1.36 | 2.89 | 36.28 | 95.5 | 81.1 | 165.32 | 24.2 | 85.53 | 0.34 | 22.95 |
| Rex | 33.42 | 1.42 | 2.58 | 38.92 | 101.40 | 77.30 | 106.70 | 20.95 | 48.40 | 0.31 | 19.00 |
| All genotypes ($n = 74$) | 29.89 | 1.20 | 2.68 | 37.84 | 121.71 | 79.72 | 191.07 | 23.60 | 67.64 | 0.27 | 21.50 |

Note:
Mean of major physiological (Photosynthesis (Pn, $\mu mol\ m^{-2}\ s^{-1}$), Stomatal conductance (Cond, $mol\ m^{-2}\ s^{-1}$), intrinsic water use efficiency (iWUE, $\mu mol\ m^{-2}\ s^{-1}/mmol$ $H_2O\ m^{-2}\ s^{-1}$), and total chlorophyll (Chl, $\mu g\ cm^{-2}$)), morphological (days to heading (DH, days), plant height (PH, cm) and shoot dry weight (SHW, g per plant)), and yield-related (panicle length (PN, cm), grain yield (GY, g per plant) and harvest index (HI, plant basis)) traits in non-stress conditions averaged across years are shown along with additive response indices (ARI) for selected genotypes.

Under non-stress, 12DS-25 and IR07F287 had the highest grain yield, which was more than twice the genotype Rex's yield. Three genotypes, namely IR08A172, IR08N136, and CT18247 out yielded Rex with better Pn, WUE, DH, SHW, GY, and HI. Most of the selected genotypes were characterized by a short growing cycle. Genotypes IR07K142 and IR07F287 had a shorter duration to flowering, accumulated high above-ground biomass and harvest index as compared to Rex and Apo. Three genotypes (12DS-25, IR07F287 and IR07K142) surpassed Apo's grain yield in the experiment. Genotypes 12DS-25, IR07F287, IR07K142 and Apo, had the highest grain yields, but their Pn values were lower by 22.4%, 21%, 18.8%, and 22% compared with the highest Pn genotype, IR08N136 (Table 3). These results indicated that higher photosynthetic efficiency cannot always increase grain yield because Pn is a complex process controlled by many genes (*Gu et al., 2014*). Among the best performers, IR08A172, IR07K142, and IR07F287 were ranked high in physiological and yield response indices (Table 3). The yield difference between the top-yielding short-duration genotype, IR07F287, and Apo was 76 g $plant^{-1}$. We identified rice genotypes with days to heading similar to Rex to explore the physiological attributes associated with high yield in the short-duration genotypes. However, IR07K142 and IR07F287 were the earliest heading/maturing genotypes with greater yield. We speculate that short-duration genotypes with higher photo assimilation during the grain-filling stage may increase yield and quality.

## CONCLUSIONS

Overall, the studied rice genotypes exhibited substantial variability for the measured growth and developmental, physiological, and yield-related traits. PCA analysis indicated that yield-related and physiological characteristics are more important for assessing the overall phenotypic variability and characterization of genotypes, followed by growth and developmental traits. Fifty-eight percent of the genotypes were classified with

moderately high and high additive index. It appears that low sink capacity during grain-filling becomes a primary cause of the antagonistic relation between long-duration and harvest index. The higher harvest index in genotypes IR07F287 and IR08A172 reflects balanced source and sink capacity. This study highlights the potential of the identified donors for developing new rice for Mid-south USA growing environments. The exploited variability and diversity could be useful for trait-based breeding. We recommend understanding short-duration genotype responses to future climatic conditions such as elevated $CO_2$, warmer temperature, and drought to solve the source and sink relationship puzzle.

## ACKNOWLEDGEMENTS

We thank the International Rice Research Institute (IRRI) for providing rice genotypes for the project. We also thank the Environmental Plant Physiology Lab staff and students for their technical help and data collection.

### Funding

This work was supported by the US Agency for International Development (USAID) through CIMMYT, the Mississippi Rice Promotion Board, and National Institute for Food and Agriculture, NIFA 2019-34263-30552 and MIS 043050. The funders had no role in study design, data collection and analysis, decision to publish, or preparation of the manuscript.

### Grant Disclosures

The following grant information was disclosed by the authors:
US Agency for International Development (USAID).
CIMMYT.
Mississippi Rice Promotion Board, and National Institute for Food and Agriculture: NIFA 2019-34263-30552 and MIS 043050.

### Competing Interests

The authors declare that they have no competing interests.

### Author Contributions

- Naqeebullah Kakar conceived and designed the experiments, performed the experiments, analyzed the data, prepared figures and/or tables, authored or reviewed drafts of the paper, and approved the final draft.
- Raju Bheemanahalli analyzed the data, prepared figures and/or tables, authored or reviewed drafts of the paper, and approved the final draft.
- Salah Jumaa performed the experiments, analyzed the data, prepared figures and/or tables, and approved the final draft.
- Edilberto Redoña analyzed the data, authored or reviewed drafts of the paper, secured the seed, and approved the final draft.

- Marilyn Warburton analyzed the data, authored or reviewed drafts of the paper, and approved the final draft.
- Kambham R Reddy conceived and designed the experiments, performed the experiments, analyzed the data, prepared figures and/or tables, authored or reviewed drafts of the paper, and approved the final draft.

### Data Availability

Raw data are available as a Supplemental File.

### Supplemental Information

Supplemental information for this article can be found online at http://dx.doi.org/10.7717/peerj.11752#supplemental-information.

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
