# Peer review of "Assessment of agro-morphological, physiological and yield traits diversity among tropical rice"

_PeerJ, doi:10.7717/peerj.11752_

## Round 0.1 · original submission · Minor Revisions

Please revise as per the comments and annotated file.

Reviewer 1 ·

Basic reporting

No comment

Experimental design

No comment

Validity of the findings

No comment

Additional comments

The information in the manuscript is really helpful for future rice breeders to plan shorter duration high yielding variety development.

Reviewer 2 ·

Basic reporting

The two references, Foley et al. (2011) and Lesk et al. (2016), are rather old to support the authors' claim that the current level of rice productivity is not enough. Consider adding a more recent reference if available. Similarly, Ray et al., (2012) is almost a ten years old reference. Is the current situation for rice yield stagnation the same as in 2012? Consider updating the reference, if possible. I highly encourage you to cite more up-to-date references when referring to agricultural statistics or the current state of farming.

In the Results section, do not repeat the numbers reported in the Tables in the text. This will hinder readability.

Figure 2: The correlation values are too small and not visible at all.

Figure 3: Define what A to H are referring to.

Figure 4: Define what A to D are referring to. Same for Figure 5.

Figure 6: Clarify what each rectangle, triangle, and circle is referring to.

Experimental design

Explain how the 74 genotypes were chosen. Were they chosen randomly or based on some criterion? This was not clear to me.

Many indices appeared in the manuscript. I did not understand the difference between additive response indices, vigor response indices, total response indices, and cumulative response indices. Or are they the same index?

I did not understand what does it mean by values > (population mean + SD), ≤ (population mean + SD) and values ≥ (population mean - SD) and < (population means - SD) in L207. I suggest rephrasing this part or provide a cartoon that explains this cutoff criterion visually.

Validity of the findings

No comment.

Additional comments

The authors evaluated 74 indica rice genotypes for morphological, physiological, and yield related phenotypes. Overall, the manuscript is well written and structured. Below are my additional comments.

L93: This sentence is a bit confusing. Did the author mean that, contrary to the rice producers in the USA, Asian rice producers do not prefer short-duration rice genotypes? If so, add a term Asia, outside of the US, or something similar.

L133: Did you mean relative humidity?

L182: RStudio is an integrated development environment for R. So you have to provide the R version the authors used for the analysis rather than the RStudio version.

L185: Panicle length was significant for the year x cultivars interaction rather than year itself. According to Table 1, the year effect is significant for spikelet fertility too.

L189: The term tolerance seems not adequate here.

L400: Remove "(".

Reviewer 3 ·

Basic reporting

Manuscript (MS) entitled «Assessment of agro-morphological, physiological and yield traits diversity among tropical rice» aimed to evaluate the variation of some traits related to yield in a wide range of tropical rice genotypes in order to identify the best individuals usable in a breeding program. It is well documented that rice is almost the main food wolrdwide but its production does not meet the demand of the growing population. Therefore, deciphering the genetic potential hidden in the germplasm is relevant for identifying elite accessions which can be used to implement a breeding program. From this point of view, the approach described in this well written MS is scientifically sound, the methodology is appropriate and significant results were recorded. Despite these significant acheivements, the paper needs slight modifications before it can be considered for publication in «Peer J».
Specific comments
Please see the attached word document of the MS.

Experimental design

The experimental was well conducted and appropriate.

Validity of the findings

The findings are relevant for the scientific community, the curator and the breeders as well.

Additional comments

Please see the attached documents.

Annotated reviews are not available for download in order to protect the identity of reviewers who chose to remain anonymous.

---

## Round 0.2 · accepted · Accept

The revisions are satisfactory.

R, in addition to Rstudio needs to be cited. The citation information can be obtained by typing "citation()" in the Rstudio console. It will be something like " R Core Team (2021). R: A language and environment for statistical computing. R Foundation for Statistical Computing, Vienna, Austria. URL https://www.R-project.org/.";